# Structure-based mechanism for activation of the AAA+ GTPase McrB by the endonuclease McrC

Neha Nirwan[1,6], Yuzuru Itoh [2,3,6], Pratima Singh[1], Sutirtha Bandyopadhyay[1], Kutti R. Vinothkumar [4,5], Alexey Amunts [2,3] & Kayarat Saikrishnan [1]

The AAA+ GTPase McrB powers DNA cleavage by the endonuclease McrC. The GTPase itself is activated by McrC. The architecture of the GTPase and nuclease complex, and the mechanism of their activation remained unknown. Here, we report a 3.6 Å structure of a GTPase-active and DNA-binding deficient construct of McrBC. Two hexameric rings of McrB are bridged by McrC dimer. McrC interacts asymmetrically with McrB protomers and inserts a stalk into the pore of the ring, reminiscent of the $\gamma$ subunit complexed to $\alpha_3\beta_3$ of $F_1$-ATPase. Activation of the GTPase involves conformational changes of residues essential for hydrolysis. Three consecutive nucleotide-binding pockets are occupied by the GTP analogue 5′-guanylyl imidodiphosphate and the next three by GDP, which is suggestive of sequential GTP hydrolysis.

[1] Division of Biology, Indian Institute of Science Education and Research, Pune 411008, India. [2] Science for Life Laboratory, Department of Biochemistry and Biophysics, Stockholm University, 17165 Solna, Sweden. [3] Department of Medical Biochemistry and Biophysics, Karolinska Institutet, 17177 Stockholm, Sweden. [4] MRC Laboratory of Molecular Biology, Cambridge CB2 0QH, UK. [5]Present address: National Centre for Biological Sciences-TIFR, GKVK Post, Bellary Road, Bangalore 560065, India. [6]These authors contributed equally: Neha Nirwan, Yuzuru Itoh. Correspondence and requests for materials should be addressed to A.A. (email: amunts@scilifelab.se) or to K.S. (email: saikrishnan@iiserpune.ac.in)

AAA+ proteins (extended ATPases associated with various cellular activities) convert chemical energy derived from nucleotide hydrolysis to mechanical form in the cell, a process fundamental to life[1,2]. A feature common to many AAA+ proteins is their activation by partner proteins thus regulating energy utilization[3–8]. McrBC is an antiphage defense system of *Escherichia coli* that specifically cleaves invading DNA that are methylated[9]. The AAA+ GTPase motor McrB is activated by the endonuclease McrC, which in turn powers DNA cleavage by the endonuclease[3]. McrBC specifically cleaves DNA containing the recognition sequence 5'-G/A(5mC)-3', where 5mC can be 5-methylcytosine, 5-hydroxymethylcytosine, or 4-methylcytosine[10,11]. Such systems have evolved to degrade methylated bacteriophage genomes that are modified as protection against other restriction-modification (RM) systems that cleave unmodified DNA[9]. They also affect horizontal gene transfer, including the spread of virulence or antibiotic resistance elements[12]. McrB is unique among AAA+ proteins as it hydrolyzes GTP rather than ATP[1]. In the presence of GTP, monomeric McrB oligomerize to hexamers[13]. Furthermore, upon binding of the endonuclease McrC, a higher oligomeric form —tetradecamers of 12 McrB and 2 McrC protomers—is formed[13]. The McrB GTPase activity is very low and is stimulated ~30-fold by McrC[3,13]. GTP hydrolysis by McrBC bound to a recognition sequence powers DNA translocation, and convergence of two such complexes on the DNA results in nucleolytic cleavage[14]. Here we present the structure of the AAA+ domain of McrB in complex with McrC at 3.6 Å resolution using cryo-electron microscopy (cryo-EM), which reveals the coupling between the endonuclease and the motor and the mechanism of activation of the GTPase. Our study provides a mechanistic framework for regulation of a AAA+ motor by its activator by remodeling of its active site.

## Results

**The dumbbell-shaped McrBΔNC.** We determined the structure of the tetradecameric complex of McrC with an McrB construct lacking the first 161 residues (McrBΔN) (Fig. 1a), which is proficient as a GTPase but does not bind DNA[13]. The complex was formed in the presence of the GTP analog 5'-guanylyl imidodiphosphate (GNP). An atomic model of McrBΔNC was built based on a 3.6 Å resolution cryo-EM map (Fig. 1b, Supplementary Figs. 1–4, Supplementary Tables 1 and 2). McrBΔNC is dumbbell shaped with the two hexameric McrBΔN rings tilted with respect to each other and sandwiching an McrC dimer (Fig. 1b, Supplementary Fig. 3). Three-dimensional (3D) classification resulted in nine major classes with varying tilt angle, indicative of conformational plasticity (Supplementary Figs. 3–5, Supplementary Movie 1). As the structure of one McrB hexamer–McrC monomer complex is identical to the other (see "Methods"), only one of them was analyzed.

**Architecture of McrBΔN.** McrBΔN has the canonical AAA+ fold comprising an N-terminal domain and a C-terminal domain (CTD) (Fig. 1c, Supplementary Fig. 6). The McrBΔN protomers form a ring-like structure, with a pseudo six-fold symmetry (Fig. 1d, Supplementary Table 3). In the hexamer, three consecutive GNP-bound interfaces have a higher buried surface than the GDP-bound interfaces (Fig. 1d–f, Supplementary Fig. 7). GDP

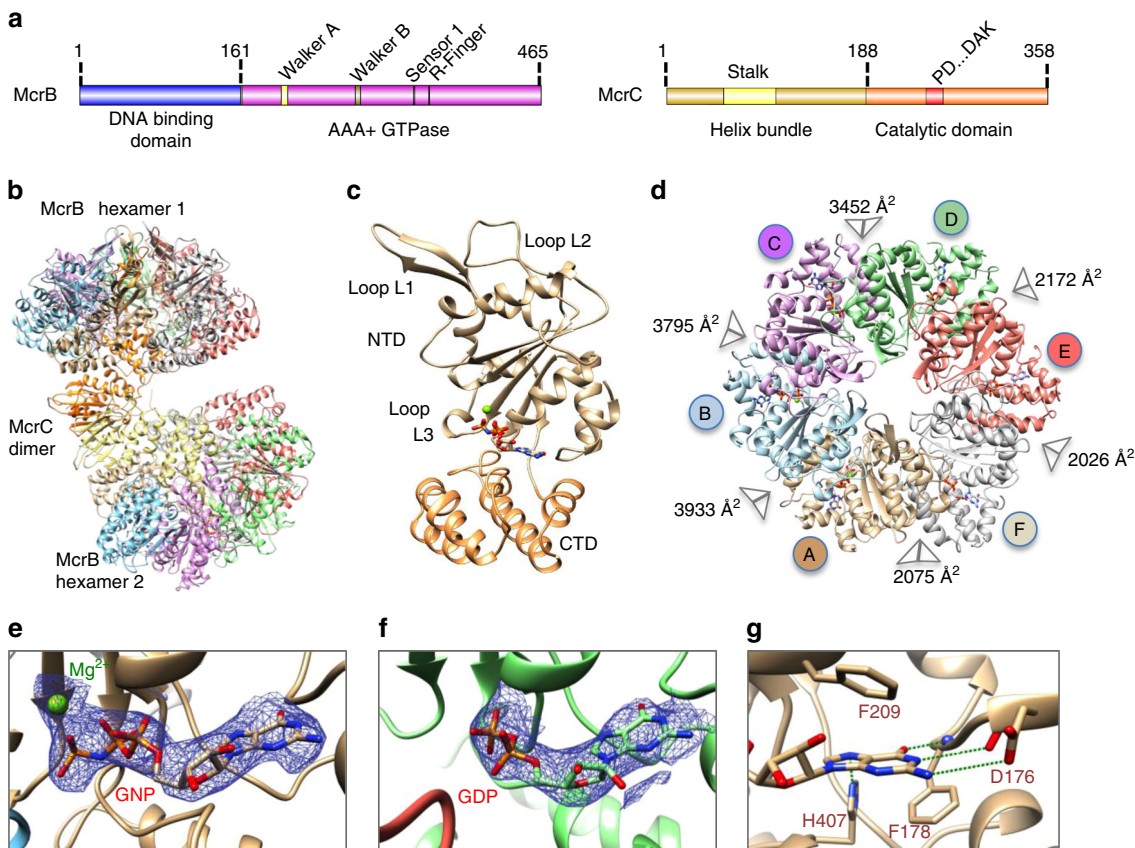

**Fig. 1** Architecture of McrBΔNC complex. **a** Domain organization of McrB and McrC. **b** Structure of McrBΔNC and **c** of an McrBΔN protomer. The bound nucleotide is shown in stick representation and the magnesium ion as a green sphere. **d** Architecture of McrBΔN hexamer. The buried surface area at the interfaces are mentioned. Cryo-electron microscopic densities shown as isosurface mesh at 1.5 σ for **e** GNP-Mg$^{2+}$ at the AB and **f** GDP at the DE interface. **g** Interactions with the guanine base at the AB interface that establish specificity for guanine. Green dashed lines represent potential hydrogen bonds

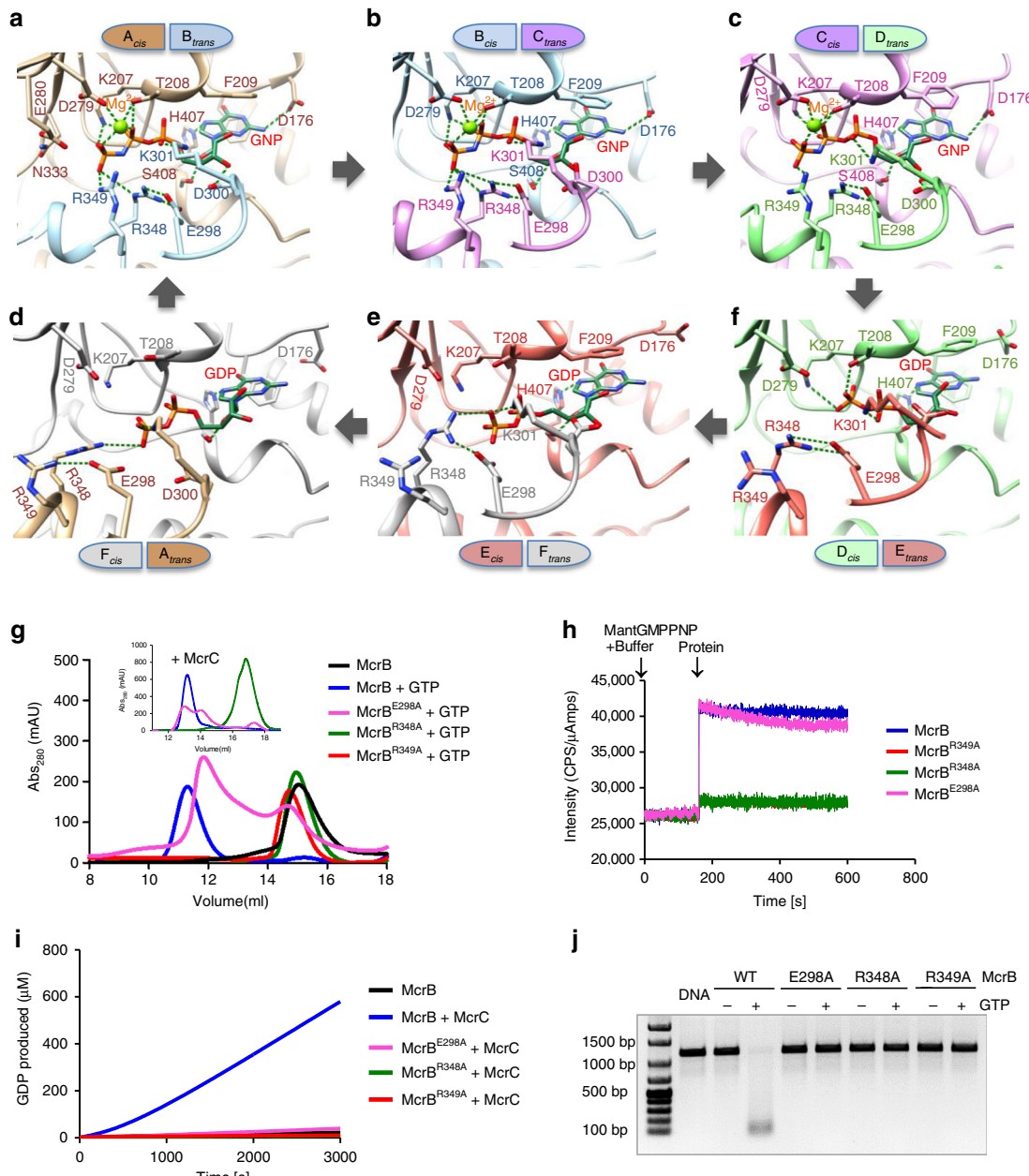

**Fig. 2** Nucleotide binding at the McrBΔN interfaces. **a–f** Six panels, arranged in a clockwise cyclic manner, illustrating the interactions made with the nucleotide at the six interfaces. Green dotted lines represent potential hydrogen bonds (<3.6 Å) and ionic interactions (<4 Å). **g** Size exclusion chromatographic profile of McrB and its mutants in the presence of GTP using Superdex 200 column and in the presence of McrC and GTP using Superose 6 column (inset). **h** Comparison of GNP binding by McrB and its mutants. The binding curves of McrB$^{R348A}$ and McrB$^{R349A}$ overlapped. **i** Comparison of the GTPase activity of McrB and its mutants in the presence of McrC. **j** Nucleolytic activity of McrB and its mutants in the presence of McrC

could have been retained during McrBΔNC purification (see "Methods"). The nucleotide-binding pocket of McrBΔN comprises the conserved Walker A, Walker B, and sensor 1 residues from one protomer (*cis*-acting) and the arginine finger from the adjacent protomer (*trans*-acting). The hydrogen bonds made by the main chain carbonyl group of McrB-Asp176$_{cis}$ and the amino group of McrB-Phe178$_{cis}$ with guanine-N1 and guanine-O6, respectively, determine the specificity for guanine (Fig. 1g). McrB-Asp176$_{cis}$ is in the vicinity of the guanine-N2 and may also contribute to base-specificity.

Depending on the bound nucleotide (GNP/GDP), the relative orientation of the protomers (Supplementary Fig. 8) and the interactions between McrBΔN and nucleotide at the six interfaces

vary (Fig. 2a–f). At the three GNP-bound interfaces, the Walker A loop interacts with the triphosphate of GNP, while Walker B McrB-Asp279$_{cis}$ coordinates the magnesium ion (Fig. 2a–c). Walker B McrB-Glu280$_{cis}$ that activates the catalytic water and the sensor 1 McrB-Asn333$_{cis}$ that orients the water[2] are near the γ-phosphate of GNP (Fig. 2a). The catalytically important arginine finger McrB-Arg349$_{trans}$ (Fig. 2g–j) interacts with the γ-phosphate.

The structure revealed that McrB does not have a canonical *cis*-acting sensor 2 that interacts with the β- and γ-phosphates. Instead, McrB-Arg348$_{trans}$, preceding the arginine finger, is within interacting distance of the phosphates. Mutation of McrB-Arg348 to alanine (McrB$^{R348A}$) affected oligomerization

and GTPase activity (Fig. 2g–j). Consequently, we propose that McrB-Arg348$_{trans}$ is the sensor 2. McrB-Glu298$_{trans}$ interacts with and orients McrB-Arg348$_{trans}$ (Fig. 2a). The importance of the interaction was highlighted by McrB[E298A]C, which bound nucleotide but formed unstable oligomers and did not hydrolyze GTP or cleave DNA (Fig. 2g–j).

The two protomers at the GDP-bound interfaces move away from each other widening the nucleotide-binding pocket, with the FA interface being the widest (Supplementary Fig. 8). At the DE interface, the dinucleotide interacts extensively with the *cis* protomer, while the *trans* protomer interacts only with the α-phosphate. In contrast, at the EF and FA interfaces only the α-phosphate interacts with the *cis*-acting Walker A, while the β-phosphate interacts with the *trans*-acting sensor 2, suggestive of stepwise loss of contact between GDP and McrBΔN. The asymmetric hexamer and the consecutive arrangement of GNP- and GDP-bound interfaces (Fig. 1d) indicated that hydrolysis of GTP occurs sequentially rather than in concerted or stochastic manner. The direction of the sequential hydrolysis is such that the transition from interface CD to DE represents the hydrolysis of GTP to GDP, while the transition from FA to AB represents exchange of the loosely bound GDP by GTP. This is consistent with the direction of hydrolysis proposed for other AAA+ proteins[15–17].

**Architecture of McrC.** The two protomers of McrC in the McrBΔNC complex are related to each other by a two-fold symmetry (Fig. 3a). Each protomer has a domain with the canonical α/β fold of the PD…D/EXK family nucleases (Fig. 3a, b, Supplementary Fig. 9). The dimeric interface mainly constitutes helix H9 and has a total buried surface area of ~3000 Å². The catalytic residues, Asp244, Asp257, and Lys259, are located close to the dimeric interface and aligned along the FA interface of McrB. The dimeric interface of McrC is similar to that of the catalytic domain of the mismatch restriction endonuclease EndoMS bound to DNA[18] (Supplementary Fig. 10). Using run off sequencing, it has previously been shown that the cleavage results in multiple nicks about ~25–33 bp away from the target site[14]. Based on the similarity with the structure of EndoMS, we speculate that the cleavage by McrBC will leave behind a 3'-overhang. The stalk's head plugs the pore of the ring through interactions with loop L1 of all the six McrBΔN protomers (Fig. 3c, d). The remaining part of the stalk interacts primarily with protomer C, mainly through van der Waals contacts. This architecture is reminiscent of the γ subunit inserting into the pore of the α₃β₃ ring of F₁-ATP synthase and regulating ATP synthesis[19].

**Asymmetric interaction of McrC remodels McrBΔN protomers.** The McrC helix bundle interacts with loop L3 (residues 333–343) and the CTD helix H9 of all the McrBΔN protomers, except F, to varying extent (Fig. 3d). These interactions change the conformation of L3. Based on conformational similarities, L3 of the six interfaces can broadly be grouped into four—that of A in one, B and C in the second, D in the third, and E and F in the fourth—as highlighted by the positions of McrB-Leu339 and McrB-Val341 within a group (Fig. 3d, Supplementary Fig. 11). Conformational change in L3 affects the *cis*-acting active site residues (Fig. 4a–f).

In comparison to L3 of protomer A$_{cis}$ at the AB interface, the conformation of L3 in the protomers B$_{cis}$ and C$_{cis}$ at BC and CD, respectively, positions the main-chain carbonyl of McrB-Leu339$_{cis}$ within hydrogen-bonding distance of McrB-Asn282$_{cis}$ (Fig. 4b, c). In addition, McrB-Asn333$_{cis}$ (sensor 1) main-chain carbonyl is hydrogen bonded to McrB-Asn282$_{cis}$, while the side

chain interacts with McrB-Glu280$_{cis}$ (Walker B). Also, the L3 residue McrB-Asp336$_{cis}$ interacts with McrB-Asn333$_{cis}$. These interactions are absent at the AB interface.

Here McrB-Asn282$_{cis}$ and McrB-Asp336$_{cis}$ appear to function as nodes that communicate the binding of McrC to McrB-Glu280$_{cis}$ and McrB-Asn333$_{cis}$. These structural alterations could activate the catalytic water for nucleotide hydrolysis (note that water molecules were not observed due to insufficient resolution). Consistent with this proposal, McrB[N282A] or McrB[D336A] are not stimulated by McrC[20]. In addition, at the CD interface, McrC-Y62 interacts with the main-chain carbonyl of McrB-Arg283$_{cis}$ of protomer C, and the region around McrB-Arg283$_{cis}$ undergoes subtle changes (Fig. 4c). The limited resolution of the map cautioned us from drawing conclusions regarding the importance of these changes to hydrolysis. Nevertheless, as the next interface, DE, is GDP-bound, we propose that at the CD interface GTP will be in a transition or hydrolyzed state.

At the DE interface, L3 of protomer D$_{cis}$ has a unique conformation that changes the position of McrB-Asp336$_{cis}$ along with McrB-Arg283$_{cis}$ (Fig. 4d). At all the GNP-bound interfaces, McrB-Arg283$_{cis}$, which appears functionally important as McrB[R283A] cannot hydrolyze GTP or cleave DNA[20], electrostatically interacts with McrB-Glu280$_{cis}$ (Fig. 4a–c). The movement of McrB-Arg283$_{cis}$ upon transition to the GDP-bound state affects its interaction with and the orientation of McrB-Glu280$_{cis}$ (Fig. 4d–f). At EF and FA interfaces, where the conformations of the *cis*-acting L3 are similar, the interaction network among the *cis*-acting active site residues is absent (Fig. 4e, f). At the next interface, AB, the *trans*-acting arginine finger and sensor 2 interact with the γ-phosphate of the GNP (Fig. 4a). The conformational change in L3 of protomer A$_{cis}$ and the sensing of the γ-phosphate re-establishes the contacts between McrB-Arg283$_{cis}$–McrB-Glu280$_{cis}$ and McrB-Asp336$_{cis}$–McrB-Asn282$_{cis}$ (Fig. 4a) through a network of interactions (Supplementary Fig. 12).

## Discussion

Based on the above analysis, we propose that the hydrolysis of GTP by McrB would be initiated at the interface that interacts with the McrC stalk (here the CD interface), while at the diametrically opposite interface (FA) a GTP would enter the hydrolysis cycle (Fig. 4g). For the cycle to continue, McrC will have to reorient such that the stalk positions at the neighboring GTP-bound interface to stimulate hydrolysis, while leaving behind a GDP-bound interface. GDP, which makes very few interactions with the interface residues, will be exchanged by GTP resulting in the movement of the protomers and formation of larger number of interactions with the interface residues. Consistent with this proposed mechanism, it has been shown previously that McrB on its own has a higher affinity for GTP than GDP[3]. The conformational changes emanating from nucleotide hydrolysis could promote the movement of McrC. A 360° rotation of McrC, akin to that of the F₁-ATP synthase γ subunit, will result in hydrolysis of six GTPs (Fig. 4g). Consequently in the tetradecamer, the dimeric McrC will appear stationary, while the two hexameric rings will rotate about their respective pivots, the McrC stalk. We note that the direction of rotation of McrC in the model is opposite to that of the γ subunit, due to the difference in the direction of nucleotide hydrolysis cycle.

Our structure-based mechanism reveals that McrC activates McrB GTPase by remodeling its active site. This is distinct from mechanisms proposed for the activation of other AAA+ proteins by their regulators. In the case of NSF, SNAP binds and alters the conformation of the domain N-terminal to the AAA+ domain to stimulate the ATPase activity[4]. Activation of the

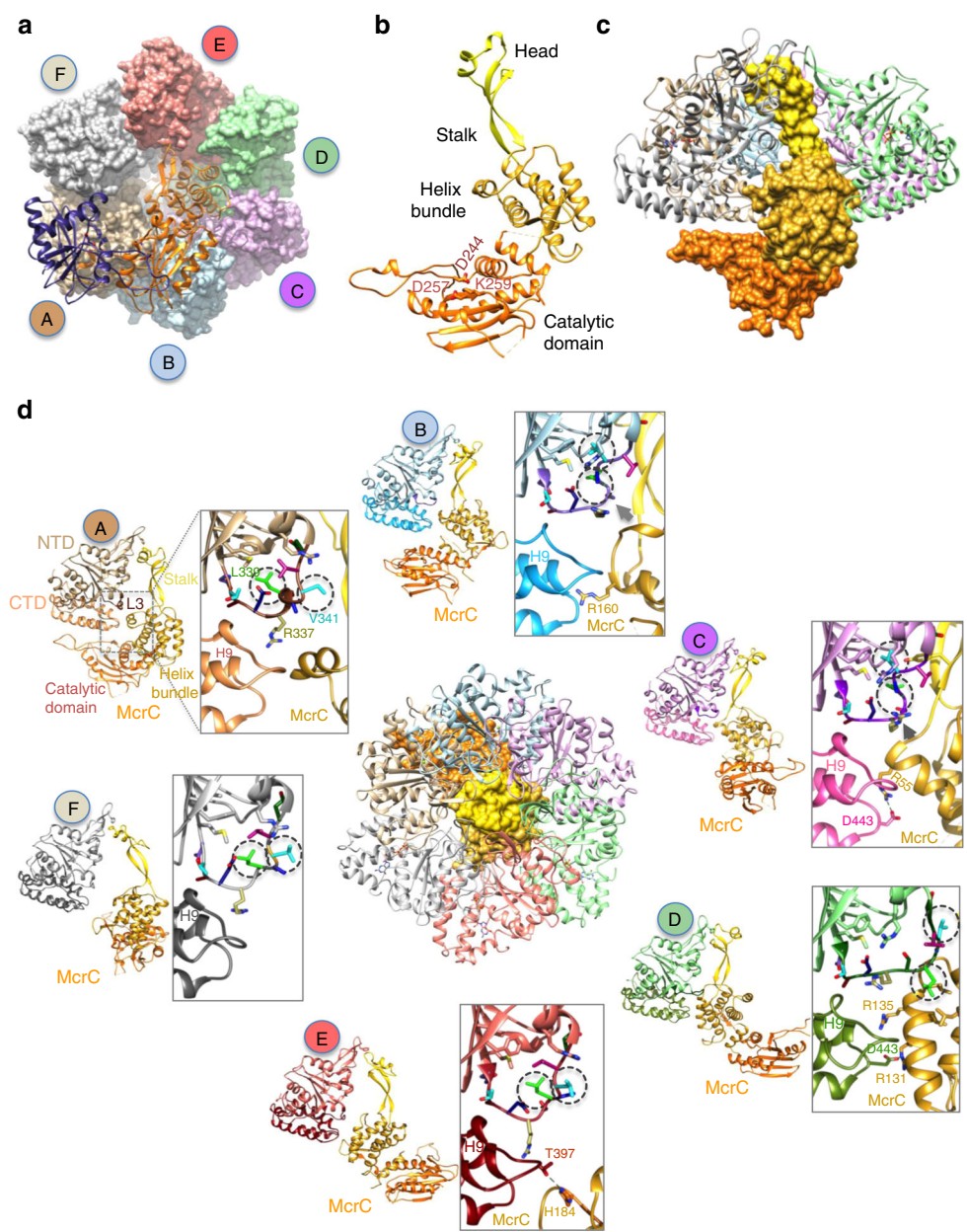

**Fig. 3** Remodeling of McrBΔN by McrC. **a** The dimer of McrC (ribbon representation) in complex with McrBΔN hexamer (surface representation). Note that this view looks at the face of the ring opposite to that in Fig. 1d. **b** The structure of an McrC monomer with the nuclease catalytic residues highlighted. **c** The asymmetric interaction of a monomer of McrC with McrBΔN hexamer. For clarity, protomer E of McrBΔN has been hidden. **d** Structure of McrBΔNC with the pore of the ring blocked by McrC is shown in the center. Around this figure, the interactions between McrC and the six McrBΔN protomers are illustrated. The inset is a zoom of the interactions between the helix bundle of McrC and L3 and C-terminal domain of McrBΔN protomers, highlighting the McrC-mediated remodeling of L3 in subunits A, B, C and D. The side chains of L3 residues are colored differently. Note the change in position of McrB-V341 (turquoise) and McrB-L339 (green), marked in dashed circles. The steric interaction between the helix bundle of McrC and L3 of B and C subunits are indicated by arrows (gray)

AAA+ ATPase Torsin, which lacks the arginine finger, by LAP1/ LULL1 is achieved by providing the arginine through hetero-oligomerization[5,6]. Thus the mechanism derived here provides a new framework for activation of AAA+ proteins, a phenomenon

that remains to be discovered and understood for many other proteins of this family.

The structure also provides mechanistic insights and poses interesting questions on the mode of DNA binding by McrBC

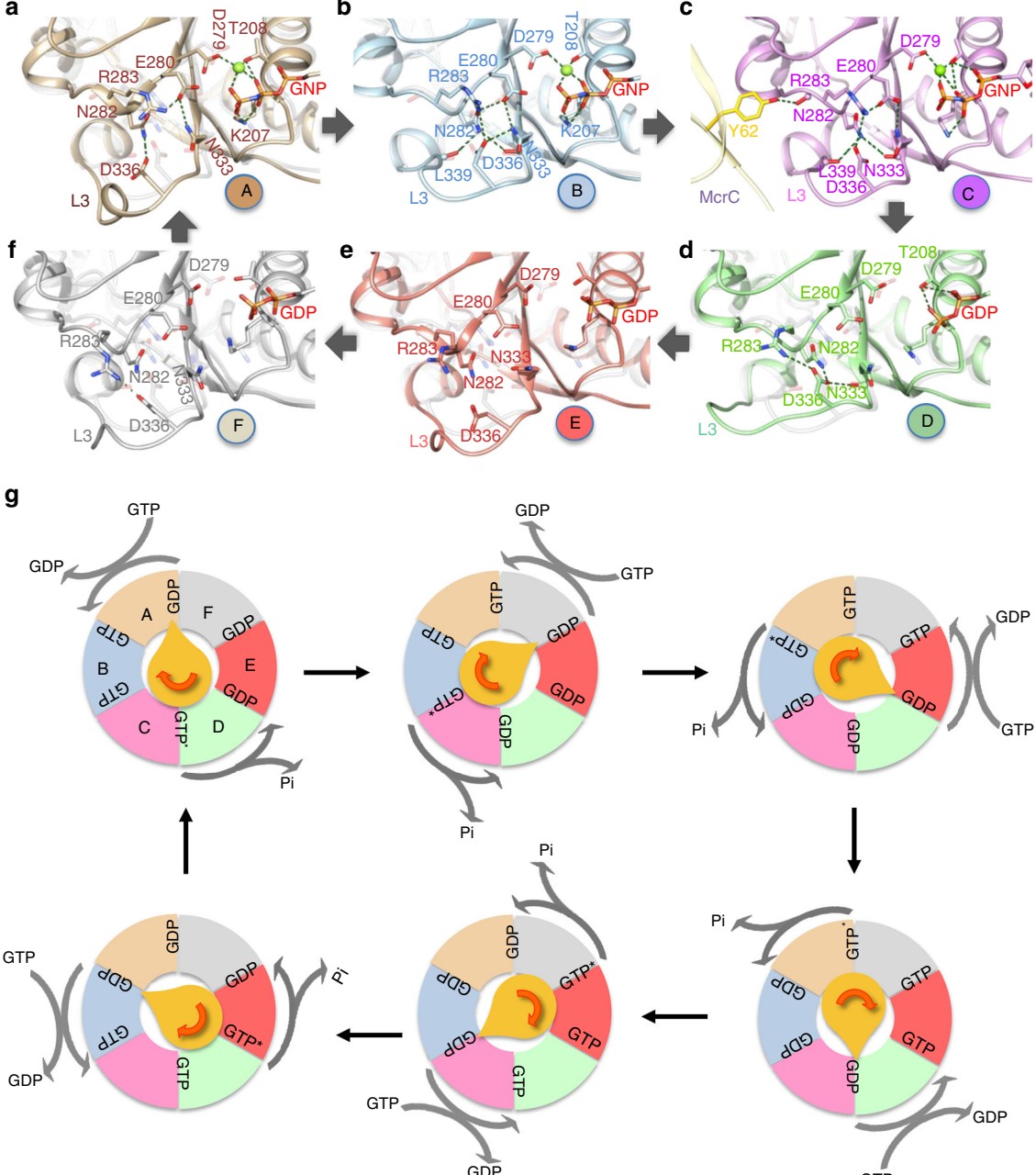

**Fig. 4** Sequential GTP hydrolysis by McrBC. **a–f** A zoom of loop L3 of McrBΔN protomers highlighting the conformational changes in them and the interaction network with the catalytically important McrB-Glu280 and Asn333. Overlaid on each panel is the faded image of this region in protomer C. The panels are arranged to follow the clockwise cyclic arrangement of the protomers in the hexamer. **g** A schematic model illustrating the McrC (yellow) stimulated GTP hydrolysis by McrB. For clarity, only one hexamer and an McrC monomer are shown, and the ring is viewed from the McrC side as in Fig. 3a. GTP* represents the transition state. The pointed end of McrC is located at the interface of GDP-to-GTP exchange and the rounded end is at the interface of GTP hydrolysis

and its nucleolytic activity. A single N-terminal DNA-binding domain of McrB recognizes and binds to the 5'-G/A(5mC)-3' target site[11]. As in the case of substrate-binding domains/subunits in other AAA+ proteins, the DNA-binding domain of McrB is likely to be located on the surface of the AAA+ hexamer and is not expected to interact with McrC[13] (Supplementary Fig. 13). Subsequent to recognition, the DNA interacts with the AAA+ domain. A common feature among nucleic acid binding and translocating AAA+ motors is threading of the substrate through the pore. The inside of the central channel of the AAA+ McrB ring shows a circular patch of positively charged residues

(Supplementary Fig. 14a), suggesting the possibility of the DNA threading through the central channel. A particularly interesting feature of McrBΔNC is that the McrC stalk blocks the pore of the McrB ring. It is possible that the DNA substrate requires sliding in through the widest interface cleft (the FA interface in the structure) or that the enzyme complex disassembles and reassembles on the substrate.

The DNA could enter the central channel of the McrB ring and fit together with the stalk of McrC. This would require rearrangement of the McrB protomers to let the DNA in. Previous studies found that DNA binding does not affect the rate of GTP

hydrolysis[3,14]. Consequently, the rearrangement of the AAA+ protomers is not expected to disturb the GTP hydrolyzing catalytic interface. However, the rearrangement could be effected at the GDP-bound interface, which has a lower interface buried surface area. Like many other AAA+ proteins bound to their DNA substrate[1], rearrangements at the GDP-bound interface could open the McrB hexamer to form a spiral/lockwasher structure and accommodate the DNA. The active site cleft of McrC with a patch of positive residues, possibly for DNA binding (Supplementary Fig. 14b), is aligned along the FA interface, suggesting a mode of engagement of the nuclease with the substrate DNA (Fig. 2a). The existing structural data does not rule out the possibility of the DNA wrapping around the external surface of the AAA+ ring, without threading into the central channel, to be fed to the active site cleft of dimeric McrC.

DNA cleavage by McrBC requires at least two recognition sequences separated by 40–3000 bp[21]. The convergence of two tetradecameric complexes possibly activates the dimeric McrC endonuclease. Cleavage happens close to one of the sites because at least one of the two tetradecamers is bound to its target site. As was proposed previously[14], and similar to Type I RM enzymes[22], the two tetradecamers could converge while remaining bound to their respective target sites via translocation-mediated DNA looping. Convergence of the two tetradecamers results in DNA cleavage by only one of them. Alternatively, one of the tetradecamers could translocate along the DNA, similar to a Type ISP RM enzyme[23], while the other remains stationery (Supplementary Fig. 15). DNA translocation will be performed by the hexameric AAA+ domains of McrB. As in the case of AAA+ helicases and proteases[1,15–17], the pore loop L1 of McrB protomers are likely to interact with the DNA asymmetrically. The sequential hydrolysis of GTP by the AAA+ domains will alter the interactions made by L1 in a cyclic manner resulting in the translocation of the DNA. We think that translocation of the DNA by McrBC tetradecamer would require engagement of the DNA with only one of the two McrB hexamer.

Interestingly, the hydrolysis of GTP does not change on DNA binding[3,14]. Hence, it is possible that the GTPase activity of McrBC is not coupled to its movement along the DNA. Instead of GTP-dependent translocation, McrBC could move along DNA by diffusion (Supplementary Fig. 14), facilitating the convergence of the two oligomers of McrBC, as in the case of Type III RM enzymes. In Type III RM enzymes, which have a Superfamily 2 helicase-like ATPase, hydrolysis of the nucleotide (ATP) makes the enzyme proficient to execute passive 1D diffusion along the DNA and cleave it[24,25]. In conclusion, the structure described here provides a platform for addressing the above questions on the mechanism of DNA cleavage by the AAA+-coupled endonuclease McrBC.

## Methods

**Mutation of McrB**. All the mutations were performed using restriction-free cloning method. The sequence of the primers used to introduce the mutations are listed in Supplementary Table 4. These PCR-amplified fragments with the mutations were used as primers in a second PCR reaction and a plasmid containing McrB wild-type gene, pHISMcrB[13], was used as the template to obtain full-length mutant genes. The amplified product was used for restriction-free cloning. All the genes were sequenced to ensure only the desired mutations were incorporated.

**Protein purification**. Proteins used for the experiment were purified using the protocol in Nirwan et al.[13]. McrB, McrBΔN, and their mutants were expressed with six histidine tags at C-terminus by overexpression of their respective plasmids (pHISMcrB, pHISMcrBΔN) while the untagged McrC was overexpressed using pHISMcrC. Cultures of transformed *E. coli* BL21(AI) cells were grown in LB broth (2 l) containing 100 µg/ml ampicillin using a shaker incubator at 37 °C to an OD$_{600}$ of 0.6. The temperature was then reduced to 18 °C, and expression was induced with 0.06% w/v L-Arabinose. The cultures were grown further overnight (15–16 h) at 18 °C. Cells were pelleted by centrifugation at 4 °C and 3315 × g for 15 min and

the pellet was resuspended in 50 ml lysis buffer (50 mM Tris-Cl pH 8, 25 mM imidazole, 500 mM NaCl, 5 mM MgCl$_2$, 10% glycerol, 0.04% CHAPS). Then the cells were lysed by sonication at 4 °C using Vibra-Cell$^{TM}$ system set. The cell lysate was then subjected to ultracentrifugation at 4 °C and 159,200 × g for 40 min in Beckman-Optima$^{TM}$ ultracentrifuge. The supernatant was then loaded onto 5 ml HiTrap$^{TM}$ Ni-NTA column (GE Life Sciences) equilibrated with Buffer A (50 mM Tris-Cl pH 8, 25 mM imidazole, 500 mM NaCl). The elution of the protein was followed by a step gradient of Buffer B (50 mM Tris-Cl pH 8, 500 mM Imidazole, 500 mM NaCl) from 5 to 100% at intervals of 20%. The purest of the Ni-NTA fractions observed in a 12% sodium dodecyl sulfate (SDS) denaturing polyacrylamide gel electrophoresis gel were dialyzed against 2 l buffer B50 (50 mM Tris-Cl pH 8, 50 mM NaCl, 1 mM EDTA, and 1 mM dithiothreitol (DTT). Dialyzed McrB or McrBΔN was loaded onto an 8 ml MonoQ 10/100 GL column (GE Life Sciences) equilibrated with Buffer B50. Protein was eluted (in 2 ml fractions) by a linear gradient of 0–50% buffer B1000 (50 mM Tris-Cl pH 8, 1000 mM NaCl, 1 mM EDTA, and 1 mM DTT) over 20 column volumes. The pure fractions based on 12% SDS denaturing gel were pooled, concentrated up to 500 µl using a 2 ml 10 kDa vivaspin2 concentrator (GE Life Sciences), and washed with 5 ml of buffer B100 (50 mM Tris-Cl pH 8, 100 mM NaCl, and 1 mM DTT) to remove EDTA. The concentrated protein was then incubated with 2.5 mM GTP and 5 mM MgCl$_2$ for 10 min at room temperature followed by centrifugation at 21,000 × g before loading onto 24 ml Superdex200 10/300 GL column (GE Life Sciences) and equilibrated with buffer B100$^{+GTP}$ (50 mM Tris-Cl pH 8, 100 mM NaCl, 0.1 mM GTP, 5 mM MgCl$_2$, 1 mM DTT). Pure fractions were pooled, concentrated, and washed with storage buffer (100 mM NaCl, 10 mM Tris-Cl pH 7.4 and 1 mM DTT) to remove GTP and finally stored in −80 °C.

McrC was purified with a method very similar to that of McrB and McrBΔN purification. After affinity purification with Ni-NTA column, to which the protein is bound, it was loaded on to 8 ml MonoS 10/100 GL column (GE Life Sciences) equilibrated with Buffer B50. Protein elution was followed by a linear gradient of buffer B1000 from 0 to 50% over 20 column volumes. Based on gel analysis, the pure fractions are pooled, washed with storage buffer, concentrated, and stored in −80 °C.

**Reconstitution of McrBΔNC complex**. After purifying the individual subunits, a complex of McrBΔN with McrC was purified using size exclusion chromatography (SEC). McrBΔN was mixed with McrC at 4-fold higher molar concentration (i.e., 4:1 ratio) and incubated with 2.5 mM GTP and 5 mM MgCl$_2$ in buffer B100 (50 mM Tris-Cl pH 8, 100 mM NaCl, 5 mM MgCl$_2$, 1 mM DTT) for 10 min at room temperature. The sample was centrifuged at 12,000 × g for 15 min before loading onto 120 ml Superdex200 10/300 GL column (GE Life Sciences), equilibrated with buffer B100 + 0.1 mM GTP. Pure fractions were pooled and concentrated using a 2-ml 10 kDa Vivaspin2 concentrator (GE Life Sciences). The concentrated protein was washed to remove GTP with a buffer containing 100 mM NaCl, 10 mM Tris-Cl pH 7.4, and 1 mM DTT. The concentrated complex was stored in storage buffer at −80 °C. The bound GDP in the structure could have resulted from the hydrolysis of GTP used during purification.

**Grid preparation**. Final concentration of 4.0 mg/ml McrBΔNC sample was prepared at room temperature in the buffer 10 mM Tris-HCl pH 7.4, 2.0 mM GNP, 0.10 M NaCl, and 5.0 mM MgCl$_2$. Three-µl of sample was applied to a freshly glow-discharged holey carbon grid (C-flat CF-2/2-4Cu-T). The grids were then incubated for 3 s at 4 °C, 100% humidity in a Vitrobot (FEI), blotted for 3 s, and plunge cooled in liquid ethane.

**Image processing**. An initial cryo-EM reconstruction of McrBΔNC at 7.4 Å was obtained using 15,918 particles subsequent to two-dimensional (2D) classification from data collected on a Titan Krios microscope (FEI) operated at 300 kV and with a Falcon 3 detector at the MRC Laboratory of Molecular Biology (Cambridge, UK). Subsequently, a larger data set was collected at the Cryo-EM Swedish National Facility in SciLifeLab (Solna, Sweden) on a Titan Krios microscope (FEI) operated at 300 kV and equipped with a K2 Summit direct electron detector (Gatan). Micrographs were obtained from automated data collections (EPU software, FEI) at ×130,000 magnification, yielding a pixel size of 1.05 Å. Eight-s exposures yielded a total dose of 30 $e^-$/Å$^2$ in 20 frames, with defocus values ranging from −0.3 to −5.0 µm. A total of 3335 micrographs were recorded and kept. Movie frames were aligned and averaged by global and local motion corrections by the program MotionCor2[26]. Contrast transfer function parameters were estimated by Gctf[27].

Particles were picked by Gautomatch and 2D classified by Relion 2[28]. In the first round, particles are picked by Gaussian-based picking, followed by reference-free 2D classification. Several representative 2D classes were used as references for the second-round picking. Finally, 771,763 particles were picked and subjected to 2D classification to discard poorly aligned particles (Supplementary Figs. 1 and 2). The remaining 761,793 particles underwent 3D classification using a 3D reference generated by Relion 3D initial model. Well-resolved classes were selected (corresponding to 225,201 particles) and subjected to 3D refinement. We tried refinement with and without C2 symmetry applied, which gave 4.1 and 3.9 Å resolution, respectively, after post processing.

To achieve higher resolution, the particles were symmetry expanded and 3D refined with local angular search applying the mask covering only one McrB hexamer and the McrC dimerizing part. The final overall resolution was 3.6 Å after post processing (Local EM map 1). To improve the local resolution of the McrB hexamer and the McrC dimerizing part, further 3D refinements with local angular search were performed applying two other local masks followed by post processing (Local EM maps 2 and 3) (Supplementary Figs. 1 and 2).

Simultaneously, 3D classification was performed to classify the movement between the McrB hexamers (Supplementary Fig. 5). Although the movement is continuous, nine distinct classes were obtained. These classes were subjected to 3D refinement and post processing. C2 symmetry was applied during refinement since it gave higher resolution than without symmetry application. Five classes gave relatively high resolution (4.2–4.8 Å, Supplementary Figs. 2–4).

**Model building and refinement**. Three locally masked maps (Local EM maps 1–3) were used for building and revising of the model using Coot. One McrB hexamer (chains A–F), a full-length McrC (chain M), and the dimerization part (residues 193–343) of the other McrC (chain N) were built as a high-resolution consensus structure. GNP, GDP, and Mg ions were placed at their binding sites. The model was subjected to energy minimization and B factor refinement against the local EM map 1 using phenix.real_space_refine in Phenix. Prior to refinement, hydrogen atoms were added to the model by ReadySet in Phenix to have better clash score. Ramachandran restraints were applied. Non-crystallographic symmetry (NCS) restrains were applied only for the dimerization parts of McrC. The refined structure was validated by MolProbity_ENREF_35[29]. The statistics are listed in Supplementary Tables 1 and 2.

The full complex models for the five relatively high-resolution 3D classes were built by superposing the refined consensus structures into the maps. The models were first rigid-body fitted in Coot[30] and further subjected to energy minimization and B factor refinement by phenix.real_space_refine. Ramachandran, NCS, and reference restraints were applied. The input models were used as the references in Phenix. The statistics are listed in Supplementary Tables 1 and 2. All the structural illustrations were made using Chimera[31].

**SEC to study oligomerization**. The oligomerization of McrB and its mutants (18 μM) in the presence and absence of McrC (4.5 μM) and GTP (2.5 mM) was studied at 4 °C using 24 ml Superdex 10/300 GL (GE Life Sciences) SEC column in the absence of McrC and 24 ml Superose6 10/300 GL SEC column (GE Life Sciences) in the presence of McrC, as described above. The equilibration buffer contained 50 mM Tris-HCl pH 8, 100 mM NaCl, 1 mM DTT, 5 mM MgCl$_2$, and 0.1 mM GTP. The injected sample volume was 400 μl.

**NADH-coupled GTPase assay**. NADH-coupled GTPase assay[32] was carried out in 200 μl reaction volume consisting of 0.8 μM of McrB or its mutants and 0.2 μM of McrC in hydrolysis buffer supplemented with 1 mM GTP (Jena Bioscience), 0.6 mM NADH (Sigma-Aldrich), 1 mM phosphoenolpyruvate, and 2 U each of pyruvate kinase and lactate dehydrogenase at 37 °C. The three enzymes were purchased from Sigma-Aldrich. The reaction was performed in 96-well flat bottom plates and absorption at 340 nm and readings were taken at an interval of 10 s for 3000 s using a Varioscan plate reader.

**DNA cleavage assay**. A 1127-bp substrate was amplified from pHISMcrBΔN plasmid by PCR using T7-Forward (5'-TAATACGACTCACTATAGGG-3') and T7-Reverse (5'-GCTAGTTATTGCTCAGCGG-3') as primers. Methylation was introduced by using 5-methyldeoxycytosine instead of deoxycytosine in the dNTP mix. The amplified product was purified from PCR mix by using the Qiagen PCR Purification Kit. Nucleolytic cleavage of DNA was carried out in 10 μl reaction mix in digestion buffer (10 mM Tris-Cl pH 8, 50 mM KCl, 5 mM MgCl2, 1 mM DTT) containing 75 ng substrate DNA incubated with 50 nM McrBC in the presence of 1 mM GTP (Jena bioscience). The reaction was incubated at 37 °C for 60 min. Two-μl 6× STES buffer (40% Sucrose, 0.2 M Tris-Cl pH 7.5, 40 mM EDTA, 1% SDS) was added and sample was heated at 65 °C for 10 min to stop the reaction. The cleaved products were resolved on a 0.8% agarose gel containing 2 μg/ml ethidium bromide at 110 V for 45 min and imaged on E-Gel$^{TM}$ Imager System (Invitrogen$^{TM}$ Life Technologies).

**Nucleotide-binding assay**. Nucleotide binding by McrB and its mutants was assessed using fluorescent analog 2'/3'-O-(N-methyl-anthraniloyl) GNP (mantGNP) (Jena Biosciences). Time-dependent change in mantGNP fluorescence was recorded on Horiba FluoroMax® 4 spectrophotometer (Jobin Yvon) at 25 °C in a 10 × 10 mm quartz cuvette. The data were collected with excitation wavelength set at 360 nm and fluorescence signal (S1/R1) was measured at 440 nm ($I_{440}$) with a slit width of 2 nm. The reaction was started by measuring the fluorescence of 400 nM mantGNP in buffer containing 10 mM Tris-Cl (pH 8.0), 50 mM KCl, 5 mM MgCl$_2$, and 1 mM DTT for 160 s followed by addition of 4 μM protein (McrB or its mutants). The fluorescence signal was further recorded till 600 s. Data from three independent experiments were averaged and plotted using GraphPad Prism 5 (GraphPad Software, Inc, San Diego, CA).

**Reporting summary**. Further information on research design is available in the Nature Research Reporting Summary linked to this article.

## Data availability

Cryo-EM maps and models are deposited in the Electron Microscopy and Protein Data Banks: EMD-0310 and 6HZ4. Maps and coordinates of five major classes are deposited — EMD-0311 and 6HZ5; EMD-0312 and 6HZ6; EMD-0313 and 6HZ7; EMD-0314 and 6HZ8; EMD-0315 and 6HZ9. Other data are available from the corresponding authors upon reasonable request.

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

## Acknowledgements

The cryo-EM data were collected at the Swedish national cryo-EM facility funded by the Knut and Alice Wallenberg, Family Erling Persson, and Kempe Foundations. We thank M. Carroni, J. Conrad, J-M de la Rosa Trevin, and S. Fleischmann for the smooth running of the data collection and processing and Manali Lalchandani for generating the clones of McrB[R348A] and McrB[R349A] mutants. K.S. acknowledges the support of Science and Engineering Research Board, Department of Science and Technology (DST), Government of India (EMR/2016/000872). N.N. acknowledges Council for Scientific and Industrial Research, India for Senior Research Fellowship. A.A and Y.I. acknowledge the support of the Swedish Foundation for Strategic Research (FFL15:0325), Ragnar Söderberg Foundation (M44/16), Swedish Research Council (NT_2015-04107), Cancerfonden (CAN 2017/1041), and H2020-MSCA-IF-2017 (799399-Itohribo). K.R.V. acknowledges the Medical Research Council grant (number U105184322) as part of Richard Henderson's group.

## Author contributions

N.N.: Conceptualization, preparation of protein samples for cryo-EM, standardization of biochemical assays, and preparation of mutant proteins and their biochemical characterization. Y.I.: Grid preparation, data collection, 3D reconstruction of high-resolution map, model building, and refinement. P.S.: SEC and GTPase assays of wild-type and mutant proteins. S.B.: GTP binding assays. K.R.V.: Grid preparation, data collection, and 3D reconstruction of the preliminary low-resolution map. A.A.: Access to cryo-EM and supervision of cryo-EM structure determination. K.S.: Conceptualization, supervision, model building, structure and data analyses, and manuscript writing with inputs from all the co-authors.

## Additional information

**Competing interests:** The authors declare no competing interests.

