## [Peer Review File · Nature Communications]

Reviewers' comments:

Reviewer #1 (Remarks to the Author):

McrBC is one of the first discovered methyl-directed restriction enzymes. So far, its studies were limited to biochemical characterization and EM experiments performed circa 1990-2000, and structural studies of the DNA-recognition domain. Recently, Nirwan et al. have shown that McrB subunits assemble into hexameric (rather than heptameric) rings, and this assembly is required for GTP hydrolysis. In the present study, Nirwan, Itoh et al. take a significant step forward, by presenting a 3.6 Å resolution cryo-EM structure of the McrBC assembly, consisting of two hexameric McrB rings bridged by an McrC dimer. The structure provides structural insight into GTP (vs ATP) specificity. But most significantly, authors show that 6 McrB subunits forming the ring differ in their interactions with neighboring subunits, bound nucleotide, and interactions with the 'stalk' structural element of McrC, which protrudes into the central (McrB)₆ channel. Based on this structural information, authors propose a model for GTP hydrolysis activation mediated by the rotation of the McrC 'stalk' relative to the McrB ring, reminiscent of the regulatory γ subunit in F₁-ATP synthase. Such mechanism of NTP hydrolysis activation is novel among AAA+ enzymes, and thus is of interest to a broad audience, making the article suitable for publication in Nature Communications.

One thing that I really miss is the actual DNA translocation mechanism, or a putative model of DNA translocation based on the available structural data. What are the possible DNA binding surfaces? Does DNA thread through the central channel of (McrB)₆, and if it does, how does it fit together with the 'stalk' of McrC? What could be the composition of the complex capable of DNA cleavage (single (McrB)₁₂(McrC)₂ complex, or two such complexes colliding)? What is the dimerization mode of the McrC PD-(D/E)XK domains, and what is the expected stagger of cleavage positions in two DNA strands? What could be the McrC activation mechanism? The present work would definitely benefit if some of the above questions were discussed in the text.

Minor points:

p2. "regulatory γ interacting" - was it "regulatory γ subunit interacting"?

p2. "McrBC specifically cleaves DNA containing the recognition sequence 5'-G/A(5mC)-3' 10,11 ." - the specificity of McrBC is broader than just 5'-G/A(5mC)-3'; it recognizes DNA with 5-methylcytosine, 5-hydroxymethylcytosine and 4-methylcytosine modifications.

p14 - the movement between the McrB hexamers is continuous. But what are the 'hinges' in the McrC subunit responsible for such movements?

p. 14. "...in Phenix to have better crash score" - was it rather "clash score"?

p25. Fig. 2h - no inset mentioned in the Fig. 2 legend is present.

p28. Fig. 4g - the McrBC stalk is represented by an almost symmetric yellow rectangle. A more visual representation would be a less symmetric shape, with edges responsible for stimulation of GDP-GTP exchange and GTP hydrolysis clearly indicated.

Reviewer #2 (Remarks to the Author):

The manuscript "Structure-based mechanism for activation of the AAA+ GTPase McrB by the endonuclease McrC" shows the AAA+ GTPase structure by X-ray crystallography and cryo-EM in which two hexameric McrB Δ N rings joined together by homodimer of McrC. The authors proposed the GTP hydrolysis mechanism, supported by the structure, in which McrC (an endonuclease) inserts a long beta-hairpin loop in the central pore of the hexameric McrB Δ N. It is a very interesting structure and I think it will create great interest to the wider scientific audience and certainly up to the level of the Nature Communication Journal.

The major drawbacks are:

1. The authors have not included the full length McrB in this study. Having seen the GTPase activity in the previous paper by the same lab, the full length McrB alone burns very little GTPs compared to full McrBC complex. It suggests that without McrC, McrB stays in an inactive state. It

would be interesting to see the low-resolution 3D model of the full length McrB alone to analyse its inactive state. It would allow an interesting comparison to the current McrB Δ NC complex structure.

2. Following the previous point, the crystal structure of the N-terminal DNA binding domain of McrB in complex with DNA has already been solved by Sukackaite et al, 2012. The McrB N-terminal domain is responsible for the recruitment and ultimately the transfer of DNA to McrC for cleavage. It would be hugely interesting to see the full length McrBC-DNA complex. It will be very informative to see which part of McrC interacts with DNA binding domain of McrB in the complex. The 2D classes and the low-resolution 3D model of the full length McrBC-DNA complex should be sufficient to analyze the overall structure of the complex.

3. This manuscript lacks the information and explanation about how 6 DNA binding domains of McrB hexamer interact with one McrC molecule.

4. In the McrBC structure, three protomers are occupied by GTP and the others with GDP. However, the asymmetry in the hexamer didn't yield an empty protomer (this observation could be more prominent in the full length McrBC-DNA complex). I believe that authors need to add some discussion about how the nucleotides are exchanged in their McrB Δ NC/McrBC structure.

Reviewer #1:

McrBC is one of the first discovered methyl-directed restriction enzymes. So far, its studies were limited to biochemical characterization and EM experiments performed circa 1990-2000, and structural studies of the DNA-recognition domain. Recently, Nirwan et al. have shown that McrB subunits assemble into hexameric (rather than heptameric) rings, and this assembly is required for GTP hydrolysis. In the present study, Nirwan, Itoh et al. take a significant step forward, by presenting a 3.6 Å resolution cryo-EM structure of the McrBC assembly, consisting of two hexameric McrB rings bridged by an McrC dimer. The structure provides structural insight into GTP (vs ATP) specificity. But most significantly, authors show that 6 McrB subunits forming the ring differ in their interactions with neighboring subunits, bound nucleotide, and interactions with the 'stalk' structural element of McrC, which protrudes into the central (McrB)₆ channel. Based on this structural information, authors propose a model for GTP hydrolysis activation mediated by the rotation of the McrC 'stalk' relative to the McrB ring, reminiscent of the regulatory γ subunit in F₁-ATP synthase. Such mechanism of NTP hydrolysis activation is novel among AAA+ enzymes, and thus is of interest to a broad audience, making the article suitable for publication in Nature Communications.

One thing that I really miss is the actual DNA translocation mechanism, or a putative model of DNA translocation based on the available structural data. What are the possible DNA binding surfaces? Does DNA thread through the central channel of (McrB)₆, and if it does, how does it fit together with the 'stalk' of McrC? What could be the composition of the complex capable of DNA cleavage (single (McrB)₁₂(McrC)₂ complex, or two such complexes colliding)? What is the dimerization mode of the McrC PD-(D/E)XK domains, and what is the expected stagger of cleavage positions in two DNA strands? What could be the McrC activation mechanism? The present work would definitely benefit if some of the above questions were discussed in the text.

Authors' response: We thank the Reviewer for taking the time to carefully read and assess the manuscript in the context of the recent developments in the AAA+ enzyme field as well as raising important points that have helped us to further discuss the functional relevance of the structure. Thanks to the Reviewer's

constructive remarks, we now have a chance to present a more thorough analysis in the revised version, including a putative model of DNA translocation and cleavage, which further highlights the results and strengthens the manuscript.

To gain insights into DNA binding, we generated an electrostatic potential surface of the hexameric McrB Δ N and McrB Δ NC, which is shown in the added Supplementary Figure 13, and extended the corresponding discussion on page 9. The interior of the central channel of the AAA+ McrB ring shows a circular patch of positive charge, indicative of complementarily charged residues that could interact with and facilitate threading of the DNA through the central channel. Importantly, the electrostatic potential surface of McrC reveals a dense patch of positive residues at the dimeric interface where the DNA is expected to bind and the active site residues of the nuclease are located. DNA could thread through the central channel of the McrB ring and fit together with the stalk of McrC in the central channel. This would require rearrangement of the McrB protomers to accommodate the DNA. Previous studies found that DNA binding does not affect the rate of GTP hydrolysis (Piper et al., 1997; Panne et al., 1999). Consequently, the rearrangement of the AAA+ protomers is not expected to disturb the GTP hydrolyzing catalytic interface. However, the rearrangement could occur at the GDP bound interface, which has a lower interface buried surface area. Like other AAA+ proteins bound to their substrate, the McrB hexamer could open to form a spiral/lockwasher structure and accommodate the DNA. The available structural data, however, does not rule out the possibility of the DNA wrapping around the external surface of the AAA+ ring without threading into the central channel, as has been proposed for DNA binding by the bacterial DnaA (Bleichert et al., 2017).

The corresponding revised lines now read as : “The structure also provides mechanistic insights and poses interesting questions on the mode of DNA binding by McrBC and its nucleolytic activity. A single N-terminal DNA binding domain of McrB recognizes and binds to the 5'-G/A(5mC)-3' target site¹¹. As in the case of substrate binding domains/subunits in other AAA+ proteins, the DNA-binding domain of McrB is likely to be located on the surface of the AAA+ hexamer, and is not expected to interact with McrC¹³ (Supplementary Figure 12). Subsequent to recognition, the DNA interacts with the AAA+ domain. A common feature amongst nucleic acid binding

and translocating AAA+ motors is threading of the substrate through the pore. The inside of the central channel of the AAA+ McrB ring shows a circular patch of positively charged residues (Supplementary Figure 13a), suggesting to the possibility of the DNA threading through the central channel. A particularly interesting feature of McrB Δ NC is that the McrC stalk blocks the pore of the McrB ring. It is possible that the DNA substrate requires to slide in through the widest interface cleft (the FA interface in the structure) or that the enzyme complex disassembles and reassembles on the substrate.

The DNA could enter the central channel of the McrB ring and fit together with the stalk of McrC. This would require rearrangement of the McrB protomers to let the DNA in. Previous studies found that DNA binding does not affect the rate of GTP hydrolysis^{3,14}. Consequently, the rearrangement of the AAA+ protomers is not expected to disturb the GTP hydrolyzing catalytic interface. However, the rearrangement could be effected at the GDP bound interface, which has a lower interface buried surface area. Like many other AAA+ proteins bound to their DNA substrate¹, rearrangements at the GDP bound interface could open the McrB hexamer to form a spiral/lockwasher structure and accommodate the DNA. The active site cleft of McrC with a patch of positive residues, possibly for DNA binding (Supplementary Figure 13b), is aligned along the FA interface, suggesting a mode of engagement of the nuclease with the substrate DNA (Figure 2A). The existing structural data does not rule out the possibility of the DNA wrapping around the external surface of the AAA+ ring, without threading into the central channel, to be fed to the active site cleft of dimeric McrC.”

Regarding the composition of the complex capable of DNA cleavage, previous studies have shown that at least two sites are required for DNA cleavage, and that the double-strand DNA break happens close to one of the two sites. It is believed that a tetradecameric complex of 12 protomers of McrB and 2 protomers of McrC binds to a target site. Hence, we think that cleavage requires convergence of two tetradecamers.

As suggested by the Reviewer, we also added a discussion about the dimerization mode of the McrC PD-(D/E)XK domains, and what is the expected stagger of cleavage positions in two DNA strands on page 5. The revised lines now read as:

“The two protomers of McrC in the McrB Δ NC complex are related to each other by a two-fold symmetry (Fig. 3a). Each protomer has a domain with the canonical α/β fold of the PD...D/EXK family nucleases (Figures 3a-b, Supplementary Fig. 9). The dimeric interface mainly constitutes helix H9, and has a total buried surface area of $\sim 3000 \text{ \AA}^2$. The catalytic residues, Asp244, Asp257 and Lys259, are located close to the dimeric interface and aligned along the FA interface of McrB. The dimeric interface of McrC is similar to that of the catalytic domain of the mismatch restriction endonuclease EndoMS bound to DNA¹⁸. Using run off sequencing it has previously been shown that the cleavage results in multiple nicks about ~ 25 to 33 bp away from the target site¹⁴. Based on the similarity with the structure of EndoMS, we predict that the cleavage by McrBC will leave behind a 3'-overhang.”

Finally, following the Reviewer's comment, we have added the analysis and discussion of the mechanism of McrC activation and DNA cleavage in the Supplementary Figure 14 and on page 10 of the revised manuscript:

“DNA cleavage by McrBC requires at least two recognition sequences separated by 40 to 3000 bp²¹. The convergence of two tetradecameric complexes possibly activates the dimeric McrC endonuclease. Cleavage happens close to one of the sites because at least one of the two tetradecamers is bound to its target site. As was proposed previously¹⁴, and similar to Type I restriction-modification (RM) enzymes²², the two tetradecamers could converge while remaining bound to their respective target sites via translocation mediated DNA looping. Convergence of the two tetradecamers results in DNA cleavage by only one of them. Alternatively, one of the tetradecamers could translocate along the DNA, similar to a Type ISP RM enzyme²³, while the other remains stationary (Supplementary Fig. 14). DNA translocation will be performed by the hexameric AAA+ domains of McrB. As in the case of AAA+ helicases and proteases^{1,15-17}, the pore loop L1 of McrB protomers are likely to interact with the DNA asymmetrically. The sequential hydrolysis of GTP by the AAA+ domains will alter the interactions made by L1 in a cyclic manner resulting

in the translocation of the DNA. Translocation of the DNA by McrBC tetradecamer would require engagement of the DNA with only one of the two McrB hexamers.

Interestingly, the hydrolysis of GTP does not change on DNA binding^{3,14}. Hence, it is possible that the GTPase activity of McrBC is not coupled to its movement along the DNA. Instead of GTP-dependent translocation, McrBC could move along DNA by diffusion (Supplementary Fig. 13), facilitating the convergence of the two oligomers of McrBC, as in the case of Type III RM enzymes. In Type III RM enzymes, which have a Superfamily 2 helicase-like ATPase, hydrolysis of the nucleotide (ATP) makes the enzyme proficient to execute passive 1D diffusion along the DNA and cleave it^{24,25}. In conclusion, the structure described here provides a platform for addressing the above questions on the mechanism of DNA cleavage by the AAA+-coupled endonuclease McrBC.”

Minor points:

p2. "regulatory γ interacting" - was it "regulatory γ subunit interacting"?

Authors' response: We have made the change.

p2. "McrBC specifically cleaves DNA containing the recognition sequence 5'-G/A(5mC)-3' 10,11 ." - the specificity of McrBC is broader than just 5'-G/A(5mC)-3'; it recognizes DNA with 5-methylcytosine, 5-hydroxymethylcytosine and 4-methylcytosine modifications.

Authors' response: We have included the other modifications that McrBC can recognize.

p14 - the movement between the McrB hexamers is continuous. But what are the 'hinges' in the McrC subunit responsible for such movements?

Authors' response: As can be observed in Supplementary Fig. 5, the McrC dimer interface is the main hinge.

p. 14. "...in Phenix to have better crash score" - was it rather "clash score"?

Authors' response: We have corrected the typographic error.

p25. Fig. 2h - no inset mentioned in the Fig. 2 legend is present.

Authors' response: We have corrected the legend for Fig. 2h.

p28. Fig. 4g - the McrBC stalk is represented by an almost symmetric yellow rectangle. A more visual representation would be a less symmetric shape, with edges responsible for stimulation of GDP-GTP exchange and GTP hydrolysis clearly indicated.

Authors' response: We have modified the figure to highlight the asymmetric structure of McrC.

Reviewer #2:

The manuscript "Structure-based mechanism for activation of the AAA+ GTPase McrB by the endonuclease McrC" shows the AAA+ GTPase structure by X-ray crystallography and cryo-EM in which two hexameric McrB Δ N rings joined together by homodimer of McrC. The authors proposed the GTP hydrolysis mechanism, supported by the structure, in which McrC (an endonuclease) inserts a long beta-hairpin loop in the central pore of the hexameric McrB Δ N. It is a very interesting structure and I think it will create great interest to the wider scientific audience and certainly up to the level of the Nature Communication Journal.

1. The authors have not included the full length McrB in this study. Having seen the GTPase activity in the previous paper by the same lab, the full length McrB alone burns very little GTPs compared to full McrBC complex. It suggests that without McrC, McrB stays in an inactive state. It would be interesting to see the low-resolution 3D model of the full length McrB alone to analyse its inactive state. It would allow an interesting comparison to the current McrB Δ NC complex structure.

Authors' response: We agree with the Reviewer that the structure of the McrB (or McrB Δ N) hexamer alone would provide additional insights into why McrB is a poor GTPase. Therefore, we have attempted cryo-EM data collection of McrB/McrB Δ N,

however preferred orientation limits the quality of that data significantly, preventing conclusive analysis. We also think that to understand the inactive or lower activity of McrB alone we would need substantially higher resolution and for this we will need to optimize the grid preparation extensively to obtain all possible views.

2. Following the previous point, the crystal structure of the N-terminal DNA binding domain of McrB in complex with DNA has already been solved by Sukackaite et al, 2012. The McrB N-terminal domain is responsible for the recruitment and ultimately the transfer of DNA to McrC for cleavage. It would be hugely interesting to see the full length McrBC-DNA complex. It will be very informative to see which part of McrC interacts with DNA binding domain of McrB in the complex. The 2D classes and the low-resolution 3D model of the full length McrBC-DNA complex should be sufficient to analyze the overall structure of the complex.

Authors' response: The attempts to investigate the full length McrBC-DNA have resulted in dissociated complex, and no new meaningful data could be obtained. We are currently analyzing the best way to stabilize and capture the complex from structural analysis. Therefore, to comply with the Reviewer comment, we generated a model using the available structural data of the N-terminal DNA binding domain and the AAA+ domain of McrB together. It appears that the DNA binding domain of McrB does not interact with McrC. This is supported by the deletion of the DNA binding domain not affecting the oligomerization of McrB with McrC or stimulation of the GTPase by McrC (Nirwan et al., 2019). Furthermore, in AAA+ proteins, such as MCM (Li et al., 2015), SV40 helicase (Gai et al., 2016), ClpB (Rizo et al., 2018) etc., whose structures have been determined with the substrate bound (DNA or peptide), the substrate binding domain is located on the upper surface of the AAA+ hexamer. In light of these observations, we propose that the DNA binding domain possibly interacts with the upper surface of the AAA+ McrB hexamer. We note that the electrostatic potential surface of the DNA binding domain has a patch of positive potential close to the C-terminus of the domain, which we found to complement a patch of negative electrostatic potential primarily contributed by residues of loop L2 of the AAA+ domain of McrB. Guided by the complementary potential surfaces, we could place the DNA binding domain on top of the AAA+ ring (Figure S12). We

modeled the structure of DNA bound to one of the six DNA binding domains based on the crystal structure of the domain obtained by Sukackaite et al. (2012). The composite structure thus obtained is reminiscent of the arrangement of the substrate binding domains/subunits in many AAA+ proteins (Supplementary Figure 12). The axis of modeled DNA aligned along the pore of the ring to be channeled in. As discussed in response to the queries of Reviewer 1, the DNA enters the central channel of the AAA+ ring from where it is fed into the active site of McrC. However, for the DNA to enter the pore without steric clash, certain protomers will have to rearrange, and, we proposed that this could be achieved if the protomers adopt a spiral/lockwasher arrangement.

3. This manuscript lacks the information and explanation about how 6 DNA binding domains of McrB hexamer interact with one McrC molecule.

Authors' response: As discussed above, we think that the six DNA binding domain of McrB hexamer do not interact with McrC.

4. In the McrBC structure, three protomers are occupied by GTP and the others with GDP. However, the asymmetry in the hexamer didn't yield an empty protomer (this observation could be more prominent in the full length McrBC-DNA complex). I believe that authors need to add some discussion about how the nucleotides are exchanged in their McrB Δ NC/McrBC structure.

Authors' response: As suggested by the Reviewer, we have discussed the mechanism of exchange of nucleotides in McrB Δ NC structure in the second paragraph of page 5:

“The two protomers at the GDP-bound interfaces move away from each other widening the nucleotide-binding pocket, with the FA interface being the widest (Supplementary Fig. 8). At the DE interface, the dinucleotide interacts extensively with the *cis* protomer, while the *trans* protomer interacts only with the α -phosphate. In contrast, at the EF and FA interfaces only the α -phosphate interacts with the *cis*-acting Walker A, while the β -phosphate interacts with the *trans*-acting sensor 2,

suggestive of stepwise loss of contact between GDP and McrB Δ N. The asymmetric hexamer, and the consecutive arrangement of GNP-bound and GDP-bound interfaces (Fig. 1d) indicated that hydrolysis of GTP occurs sequentially rather than in concerted or stochastic manner. The direction of the sequential hydrolysis is such that the transition from interface CD to DE represents the hydrolysis of GTP to GDP, while the transition from FA to AB represents exchange of the loosely bound GDP by GTP. This is consistent with the direction of hydrolysis proposed for other AAA+ proteins¹⁵⁻¹⁷.”

We have discussed this further in page 8 of the revised manuscript as follows:
“GDP, which makes very few interactions with the interface residues, will be exchanged by GTP resulting in the movement of the protomers and formation of larger number of interactions with the interface residues. Consistent with this proposed mechanism, it has been shown previously that McrB on its own has a higher affinity for GTP than GDP³.”

REVIEWERS' COMMENTS:

Reviewer #1 (Remarks to the Author):

In the revised version authors have updated and expanded the discussion related to the McrBC-DNA interactions, and they also propose possible mechanisms for McrBC action.

This work merits publication in Nature Communications, given the minor points related to the expanded discussion and the extra figures included in the revised version are addressed:

Supplementary Fig. 12. Left-side image depicts the McrB Δ N hexamer with a single DNA-bound N-domain. But it is not clear what is the linker length between the C-terminus of the X-ray McrB-N structure and the N-terminus of McrB Δ N. The linker attachment sites should be shown, and the length of the unshown linker in aa stated in the legend. The right-side image apparently depicts a model of full-length McrB hexamer with a single DNA bound, including the linkers. The linkers should be marked and the figure legend updated to clearly state the presence of 6 McrB-N domains (vs just a single McrB-N domain in the left-side image).

Supplementary Fig. 13. Authors state in the revised text that the pore formed by the McrB Δ N hexamer is of sufficient size to accommodate dsDNA even in the presence of the McrC 'stalk'. However, it is not clear from the presented figures what is the actual pore diameter in the McrB ring, and what part of this diameter is occluded by the McrC.

Supplementary Fig. 14. The authors depict putative reaction mechanism for McrBC, based on the established mechanisms for other restriction-modification enzymes (Type I-like NTP-driven translocation or Type III-like NTP-triggered diffusion) and the McrBC mechanism proposed by Panne et al 1999. The McrC activation/cleavage occur upon collision of two McrBC 14-subunit complexes (one stationary and one mobile) close to the recognition site with the stationary 14-subunit complex. But this model does not account for the McrBC cleavage pattern reported by Pieper et al. (Biochemistry 2002, 41, 5245-5254), i.e. multiple cleavage positions with an apparent ~10 bp repeat. Cleavage at multiple positions between the recognition sites would require simultaneous movement of both 14-subunit complexes away from the recognition sites (the recognition site could remain bound to the McrB-N or released as in Fig. s14), and collision at a random position between the sites.

Alternatively, a similar translocation/movement & cleavage could be accomplished by a single 14-subunit complex, with McrB-N domain from the first McrB hexamer interacting with the first recognition site, the McrB-N domain from the second McrB hexamer interacting with the second recognition site, and the intra-site DNA forming a loop. This loop could be subsequently reduced by the DNA translocation activity (and, possibly, extrusion of 2 extra loops) by the McrB hexamers. Though it is not possible to discriminate between these models based on available data, such models could be included as extra supplementary material.

Supplementary Fig. 5. In the reply to the question about the 'hinges' in McrC responsible for multiple orientations of McrB hexamers, authors state 'As can be observed in Supplementary Fig. 5, the McrC dimer interface is the main hinge.'. But to me it seems that the McrC dimers in both pink and yellow classes presented in Fig. S5 overlap quite well, and the major difference is the position of one McrB hexamer relative to the McrC dimer. This implies that the 'hinge' responsible for the movement is somewhere between the globular part of McrC and the 'stalk'. Otherwise, if the dimer interface of McrC indeed adopts multiple states, this highly unusual mode of PD-EXK domain dimerization should be discussed in detail.

In the reply to the question about the McrC dimerization mode / overhangs, authors state that the McrC dimer is most similar to EndoMS endonuclease, implying a 3'-overhang left upon dsDNA cleavage. A figure showing the overlay between EndoMS and (McrC)₂ would be helpful.

Reviewer 1 comments:

In the revised version authors have updated and expanded the discussion related to the McrBC-DNA interactions, and they also propose possible mechanisms for McrBC action.

This work merits publication in Nature Communications, given the minor points related to the expanded discussion and the extra figures included in the revised version are addressed:

Authors' response: We are grateful to the Reviewer for taking the time to read the revised manuscript carefully and for the comments to further improve the manuscript. Please find below our response to the comments point-by-point.

Supplementary Fig. 12. Left-side image depicts the McrB Δ N hexamer with a single DNA-bound N-domain. But it is not clear what is the linker length between the C-terminus of the X-ray McrB-N structure and the N-terminus of McrB Δ N. The linker attachment sites should be shown, and the length of the unshown linker in aa stated in the legend. The right-side image apparently depicts a model of full-length McrB hexamer with a single DNA bound, including the linkers. The linkers should be marked and the figure legend updated to clearly state the presence of 6 McrB-N domains (vs just a single McrB-N domain in the left-side image).

Authors' response: We have replaced the two figure panels with new ones to highlight the C-terminal and the N-terminal ends of McrB-N and McrB Δ N, respectively in Supplementary Fig. 13 of the revised manuscript (Supplementary Fig. 12 in the previous version). The legend has been revised to indicate the length of the linker connecting the two domains, and state the difference between the figures in the two panel.

Supplementary Fig. 13. Authors state in the revised text that the pore formed by the McrB Δ N hexamer is of sufficient size to accommodate dsDNA even in the presence of the McrC 'stalk'. However, it is not clear from the presented figures what is the actual pore diameter in the McrB ring, and what part of this diameter is occluded by the McrC.

Authors' response: The pore of the ring is not sufficiently large to allow the passage of the DNA. We have clarified this in the legend to Supplementary Fig. 13 of the revised manuscript (Supplementary Fig. 12 in the previous version). Also, the shaft of McrC occludes the pore almost completely (Fig. 3). We have also mentioned that to accommodate the DNA along with the shaft the protomers will have to rearrange. This has been discussed in the manuscript (Page 10).

Supplementary Fig. 14. The authors depict putative reaction mechanism for McrBC, based on the established mechanisms for other restriction-

modification enzymes (Type I-like NTP-driven translocation or Type III-like NTP-triggered diffusion) and the McrBC mechanism proposed by Panne et al 1999. The McrC activation/cleavage occur upon collision of two McrBC 14-subunit complexes (one stationary and one mobile) close to the recognition site with the stationary 14-subunit complex. But this model does not account for the McrBC cleavage pattern reported by Pieper et al. (Biochemistry 2002, 41, 5245-5254), i.e. multiple cleavage positions with an apparent ~10 bp repeat. Cleavage at multiple positions between the recognition sites would require simultaneous movement of both 14-subunit complexes away from the recognition sites (the recognition site could remain bound to the McrB-N or released as in Fig. s14), and collision at a random position between the sites. Alternatively, a similar translocation/movement & cleavage could be accomplished by a single 14-subunit complex, with McrB-N domain from the first McrB hexamer interacting with the first recognition site, the McrB-N domain from the second McrB hexamer interacting with the second recognition site, and the intra-site DNA forming a loop. This loop could be subsequently reduced by the DNA translocation activity (and, possibly, extrusion of 2 extra loops) by the McrB hexamers. Though it is not possible to discriminate between these models based on available data, such models could be included as extra supplementary material.

Authors' response: As suggested by the Reviewer, we have included other possible models of DNA cleavage in the legend for Supplementary Fig. 15 of the revised manuscript (Supplementary Fig. 14 in the previous version).

Supplementary Fig. 5. In the reply to the question about the 'hinges' in McrC responsible for multiple orientations of McrB hexamers, authors state 'As can be observed in Supplementary Fig. 5, the McrC dimer interface is the main hinge.'. But to me it seems that the McrC dimers in both pink and yellow classes presented in Fig. S5 overlap quite well, and the major difference is the position of one McrB hexamer relative to the McrC dimer. This implies that the 'hinge' responsible for the movement is somewhere between the globular part of McrC and the 'stalk'. Otherwise, if the dimer interface of McrC indeed adopts multiple states, this highly unusual mode of PD-EXK domain dimerization should be discussed in detail.

Authors' response: From the structures of McrB Δ NC obtained from the different map classes, the main hinge appears to be the dimer interface. This is clear from the figure below. However, based on this analysis, we cannot rule out the possibility of other hinges in McrC.

Figure legend: Superposition of one of the protomers of McrC obtained from the structure of class 4 on to the corresponding protomer class 1. The superposition highlights the variation in the dimer interface serving as the hinge.

The difference in the dimer interface is not very large and indicates the plasticity of the interface. Such variations have been noted in case of other endonucleases too. For example, in the case of EndoMS, the catalytic domains do not dimerize in the apo form. It is possible that the dimer interface of McrC rigidifies on binding to DNA. As the variation in dimer interface is small and not unusual, we have not discussed it in the manuscript.

In the reply to the question about the McrC dimerization mode / overhangs, authors state that the McrC dimer is most similar to EndoMS endonuclease, implying a 3'-overhang left upon dsDNA cleavage. A figure showing the overlay between EndoMS and (McrC)₂ would be helpful.

Authors' response: As suggested by the Reviewer, we have included a figure showing the superposition of the catalytic domain of McrC and EndoMS endonuclease (Supplementary Fig. 10 in the revised manuscript).